# PHYSICS-INFORMED NEURAL NETWORKS WITH UNKNOWN MEASUREMENT NOISE

## ABSTRACT

Physics-informed neural networks (PINNs) constitute a flexible approach to both finding solutions and identifying parameters of partial differential equations. Most works on the topic assume noiseless data, or data contaminated by weak Gaussian noise. We show that the standard PINN framework breaks down in case of non-Gaussian noise. We give a way of resolving this fundamental issue and we propose to jointly train an energy-based model (EBM) to learn the correct noise distribution. We illustrate the improved performance of our approach using multiple examples.

## 1 INTRODUCTION

While the idea of using neural networks to solve partial differential equations (PDEs) dates back to the work of Lagaris et al. (1998), the field has received renewed attention (Cuomo et al., 2022; Karniadakis et al., 2021; Markidis, 2021; Blechschmidt and Ernst, 2021) due to the seminal work of Raissi et al. (2019), in which they introduced physics-informed neural networks (PINNs). Both the forward problem, where the solution to a PDE is learned given the boundary conditions, as well as the inverse problem, where parameters of the PDE are to be inferred from measurements, can be solved with the PINN approach.

A multitude of applications for PINNs in science have already been considered: Cai et al. (2021a) review the application of PINNs to fluid mechanics, Cai et al. (2021b) review their application to heat transfer problems, Yazdani et al. (2020) utilize PINNs to infer parameters for systems of differential equations in systems biology, Mao et al. (2020) investigate the applicability of PINNs to high-speed flows, and Sahli Costabal et al. (2020) utilize PINNs to take into account wave propagation dynamics in cardiac activation mapping.

The main advantage of PINNs over traditional solvers lies in their flexibility, especially when considering the inverse problem (Karniadakis et al., 2021): as neural networks, they have the capacity for universal function approximation, they are mesh-free, and they can directly be applied to very different kinds of PDEs, without the need to use a custom solver. When dealing with the inverse problem, parameters are learned from data and the question as to the effect of noisy data on the quality of the estimates arises naturally. However, most of the existing work on PINNs assumes either noiseless data, or data contaminated with weak Gaussian noise. While some research has been done on the effects of noisy data in PINN training (see Section 2), they still consider either Gaussian noise or are focused on uncertainty quantification.

In this work, we consider the inverse problem for the case of measurements contaminated by non-Gaussian noise of unknown form. The least-squares loss, which is commonly employed as data loss in PINNs, is known to perform poorly in this case (Constable, 1988; Akkaya and Tiku, 2008). We give a way of mitigating this issue by suitably modeling the noise distribution. A high-level illustration of our method is given in Fig. 1: we employ an energy-based model (EBM) to learn the noise probability density function (PDF) jointly with the PINN. This PDF is then utilized to estimate the likelihood of the measurements under our model, which in turn serves as data loss.

Non-Gaussian measurement noise of unknown form can appear in a variety of applications (Johnson and Rao, 1991). Measurements of geomagnetic fields may exhibit asymmetric, long-tailed noise (Constable, 1988), data in astrophysics or seismology may be contaminated with impulsive noise (Weng and Barner, 2005), and not accounted for systematic errors in the measurement procedure

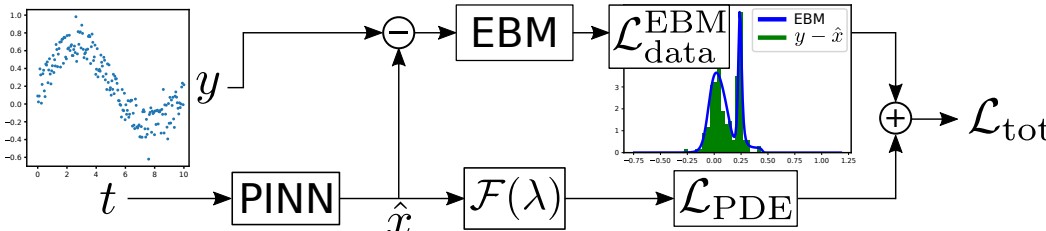

Figure 1: The PINN-EBM approach: the inputs $t$ are passed through the PINN to obtain the PINN predictions $\hat{x}$. The differential operator $\mathcal{F}(\lambda)$ is then applied to $\hat{x}$ to obtain the PDE residuals, which are subsequently utilized to calculate the PDE loss $\mathcal{L}_{\text{PDE}}$. At the same time, the residuals between the noisy measurements $y$ and $\hat{x}$ are formed, serving as noise estimates. The EBM is then trained on these estimates in order to learn the noise PDF, which can in turn be utilized to compute the likelihood of the measurements serving as data loss $\mathcal{L}_{\text{data}}^{\text{EBM}}$. Finally, the two loss terms are combined to form the total loss $\mathcal{L}_{\text{tot}}$. Both PINN and EBM, as well as the PDE parameters $\lambda$, can be trained by backpropagating $\mathcal{L}_{\text{tot}}$.

may give rise to bias in the noise (Barlow, 2002). For example, the noise distribution considered in Fig. 1 can be interpreted as resulting from the case where measurements of multiple sensors have been merged into one dataset. One of the sensors produces biased measurements, giving rise to the second peak away from zero. Then the PINN-EBM allows for solving the PDE correctly together with the simultaneous detection of previously unrecognized systematic errors in the data.

## 2 RELATED WORK

Some research has been done on PINNs in case of noisy measurements, although typically only Gaussian noise is considered. In Yang et al. (2021), the framework of Bayesian neural networks (Goan and Fookes, 2020) is combined with the PINN framework, in order to obtain uncertainty estimates when training PINNs on noisy data. Bajaj et al. (2021) introduce the GP-smoothed PINN, where a Gaussian process (GP) (Rasmussen and Williams, 2006) is utilized to ameliorate noisy initial value data and make the PINN training more robust in this situation. In Chen et al. (2021), the PINN-SR method is introduced which can be employed to determine governing equations from scarce and noisy data. In contrast to these works, we give a way of taking into account unknown, non-Gaussian noise in the PINN and provide a training procedure for this case.

While EBMs are commonly employed for classification (LeCun et al., 2006) and image generation (Du and Mordatch, 2019), they have also been successfully applied to regression problems; applications include object detection (Gustafsson et al., 2020a) and visual tracking (Danelljan et al., 2020). In Gustafsson et al. (2020b), different methods for training EBMs for regression are discussed. In our work we also consider regression tasks and to the best of our knowledge, our paper is the first to combine EBMs and PINNs. The standard EBM approach to regression would not take the physical knowledge in form of the differential equation into account.

## 3 BACKGROUND

In this section we give a brief introduction to the two distinct methods which we will combine in our work: physics-informed neural networks and energy-based models.

### 3.1 PHYSICS-INFORMED NEURAL NETWORKS (PINNS)

The PINN-framework (Raissi et al., 2019) can be used as an alternate approach to solving PDEs, other than the standard, mesh-based solvers. The objective is to numerically determine the solution to a differential equation $\mathcal{F}x(t) = 0$, where $\mathcal{F}$ denotes the differential operator defining the PDE, $x(t)$ the solution to the differential equation, and $t$ the input; both $x$ and $t$ can be multidimensional. In the PINN approach, we now employ a neural network to parameterize the numerical solution

$\hat{x}(t) = \hat{x}(t|\theta_{\text{PINN}})$ of the PDE, where $\theta_{\text{PINN}}$ denotes the weights of the PINN which are to be optimized.

In this paper, we consider the inverse problem: we are given a dataset $\mathcal{D}_d = \{t_d, y_d\}$ of $N_d$ (noisy) measurements of the PINN solution in addition to the parametric form of the differential operator $\mathcal{F}(\lambda)$, which may contain unknown parameters $\lambda$. Furthermore, we choose another set of $N_c$ so-called collocation points $\mathcal{D}_c = \{t_c\}$. They can lie at arbitrary points in the input domain of the PDE and are used to take into account the PDE constraint. As long as collocation points are placed in the areas of interest, they can enable the PINN to extrapolate also to areas without measurements.

When training the PINN, two losses enter into the loss function: the data loss, $\mathcal{L}_{\text{data}}$, evaluating the fit of the PINN prediction with the data, and the PDE loss, $\mathcal{L}_{\text{PDE}}$, a measure of the fulfillment of the PDE by the PINN solution. In the standard PINN, the least-squares loss is used as data loss,

$$\mathcal{L}_{\text{data}}(\hat{x}, \{t_d, y_d\}_{\text{mb}}) = \frac{1}{N_d'} \sum_{i=1}^{N_d'} (\hat{x}(t_d^i) - y_d^i)^2, \tag{1}$$

and the squares of the PDE residuals, $f(t) = \mathcal{F}(\lambda)\hat{x}(t) \overset{!}{=} 0$, at the collocation points serve as PDE loss,

$$\mathcal{L}_{\text{PDE}}(\mathcal{F}, \hat{x}, \{t_c\}_{\text{mb}}) = \frac{1}{N_c'} \sum_{i=1}^{N_c'} f(t_c^i)^2. \tag{2}$$

Here, $N_d'$ and $N_c'$ denote the number of data points $t_d$ and collocation points $t_c$, respectively, in the current mini-batch (mb). The PINN is then trained by minimizing the total loss $\mathcal{L}_{\text{tot}} = \mathcal{L}_{\text{data}} + \omega\mathcal{L}_{\text{PDE}}$ with respect to the parameters $\theta_{\text{PINN}}$ and $\lambda$. Unless stated otherwise, the weighting factor $\omega = 1$. Utilizing this loss function, the PINN is optimized via some variation of gradient descent.

## 3.2 Energy-based models (EBMs)

EBMs constitute a powerful method of learning probability densities from data (LeCun et al., 2006). They are frequently employed for image generation and modeling (Du and Mordatch, 2019; Nijkamp et al., 2020), and have successfully been applied to the domain of regression (see Section 2). The EBM can retain all of the flexibility of a neural network, via the following parametrization:

$$p(y|t, \hat{h}) = \frac{e^{\hat{h}(t,y)}}{Z(t, \hat{h})}, \qquad Z(t, \hat{h}) = \int e^{\hat{h}(t,\tilde{y})} d\tilde{y}, \tag{3}$$

where $\hat{h}(t, y) = \hat{h}(t, y|\theta_{\text{EBM}})$ is the (scalar) output of the neural network with weights $\theta_{\text{EBM}}$.

This expressive capacity, however, comes at the cost of a more complicated training procedure, since the partition function $Z(t, \hat{h})$ will typically be analytically intractable. In case of a high-dimensional $y$, it will also be expensive or infeasible numerically; Monte Carlo methods are often employed to approximate the intractable integral in (3). Various ways of training EBMs for regression are given in Gustafsson et al. (2020b).

In this paper, we will choose the approach of directly minimizing the negative log-likelihood (NLL), which can be written as

$$\text{NLL}(\{t_d, y_d\}, \hat{h}) = -\log\left(\prod_{i=1}^{N_d} p(y_d^i|t_d^i, \hat{h})\right) = \sum_{i=1}^{N_d} \log Z(t_d^i, \hat{h}) - \hat{h}(t_d^i, y_d^i). \tag{4}$$

In our case, the evaluation of the partition function to high accuracy remains tractable by utilizing numerical integration, since the noise distribution is one-dimensional.

## 4 Problem formulation

We consider the following setup, consisting of the differential equation and a measurement equation:

$$\mathcal{F}(\lambda)x(t) = 0, \qquad y(t) = x(t) + \epsilon, \tag{5}$$

where the parametric form of the differential operator $\mathcal{F}(\lambda)$ is given, as well as $N_d$ measurements $y(t)$ of the corresponding solution $x(t)$ contaminated with homogeneous measurement noise $\epsilon$. In this work, we aim to combine the PINN and the EBM framework in order to solve the inverse problem in case of measurements contaminated with non-Gaussian and non-zero mean noise.

## 4.1 PINNs in case of non-zero mean noise

To see why noise with non-zero mean is problematic in the standard PINN framework, consider the loss function $\mathcal{L}_{\text{tot}} = \mathcal{L}_{\text{data}} + \omega \mathcal{L}_{\text{PDE}}$ (compare Section 3.1):

$$\mathcal{L}_{\text{tot}} = \frac{1}{N_d'} \sum_{i=1}^{N_d'} \left( \hat{x}(t_d^i) - y_d^i \right)^2 + \frac{\omega}{N_c'} \sum_{i=1}^{N_c'} f(t_c^i)^2. \tag{6}$$

A major challenge when training PINNs, even in case of Gaussian noise, lies in the fact that data and PDE loss counteract each other, since the first term would encourage overfitting towards each data point, whereas the second term would push the solution towards fulfilling the PDE.

Let us first consider the case of a zero-mean noise distribution in the limit of infinite data generated according to (5), and assume that the PINN has sufficient capacity to represent the solution accurately. Then both loss terms would be minimal for $\hat{x}(t) = x(t)$: in the limit, the data loss would become $\lim_{N_d \to \infty} \mathcal{L}_{\text{data}} = \mathbb{E}[(\hat{x} - x - \epsilon)^2] = \mathbb{E}[(\hat{x} - x)^2] + \text{Var}_\epsilon$, and hence $\hat{x} = x$. The PDE loss would then obviously also be minimal (more precisely zero) for $\hat{x} = x$, due to (5).

For noise with non-zero mean, however, the problem of mismatch between the two loss terms is not simply an effect of finite data, which could be treated via an adequate regularization procedure, but has deeper roots. Considering again the limit of infinite data, we have $\lim_{N_d \to \infty} \mathcal{L}_{\text{data}} = \mathbb{E}[(\hat{x} - x - \epsilon)^2] = \mathbb{E}[(\hat{x} - x - \mu_\epsilon)^2] + \text{Var}_\epsilon$, and the minimizer of the data loss becomes $\hat{x} = x + \mu_\epsilon$, where $\mu_\epsilon$ denotes the mean of the noise distribution. For the PDE loss, on the other hand, the minimizer would remain $\hat{x} = x$ (assuming that the correct parameters $\lambda$ of $\mathcal{F}(\lambda)$ are known).

The two losses are now counteracting each other and the optimization procedure will produce some compromise between them, even when the parameters $\lambda$ of the PDE are known exactly a priori. In the case where the parameters of $\mathcal{F}(\lambda)$ are unknown, the optimization procedure will tend to converge to parameter values which result in a lower least-squares loss than the correct ones might.

## 4.2 Reconciling the losses

A simple way of restoring consistency between the two losses in the limit of infinite data is to add an offset parameter $\theta_0$ to the PINN prediction $\hat{x}$ in the data loss term, which is supposed to learn the bias in the noise term. The data loss then reads $\lim_{N_d \to \infty} \mathcal{L}_{\text{data}} = \mathbb{E}[(\hat{x} + \theta_0 - x - \epsilon)^2] = \mathbb{E}[(\hat{x} + \theta_0 - x - \mu_\epsilon)^2] + \text{Var}_\epsilon$ and the optimum $\hat{x} = x, \theta_0 = \mu_\epsilon$ would be compatible with the PDE loss. Hence the optimization procedure could converge to the correct solution, in principle.

However, since the maximum likelihood estimator is asymptotically optimal only when using the correct likelihood (Wasserman, 2004), this data loss still does not constitute the best option (except for the case of Gaussian noise with non-zero mean). Furthermore, in the practical case of finite data, outliers may have an outsized effect when using the least-squares loss. Hence, a way of taking the non-Gaussianity of the noise into account explicitly would likely further improve the speed of optimization as well as the final learning outcome. In the next section, we demonstrate how EBMs can be used for this purpose.

## 4.3 Using EBMs to learn the noise distribution

In order to also take the shape of non-Gaussian noise into account, we can choose to use a different data loss in (6). A natural choice would be the log-likelihood of the measurements, given the noise distribution. In case of Gaussian noise, this would again result in a least-squares loss term, plus a constant. However, since in our problem setup we do not know the form of the noise a priori, we need to determine the shape of the noise distribution jointly with the PINN solution.

---

**Algorithm 1:** Training the PINN-EBM

---

**Input** : PINN $\hat{x}(\cdot|\theta_{\text{PINN}})$, EBM $\hat{h}(\cdot|\theta_{\text{EBM}})$, data points $\{t_d, y_d\}$, collocation points $\{t_c\}$,
differential operator $\mathcal{F}(\lambda)$, weighting factor $\omega$
**Output:** optimized $\theta_{\text{PINN}}, \theta_{\text{EBM}}, \lambda$
**while** *Training* **do**
    Draw a minibatch of data points $\{t_d, y_d\}_{\text{mb}}$ and of collocation points $\{t_c\}_{\text{mb}}$
    **if** $i < i_{\text{ebm}}$ **then**
        | Calculate data loss $\mathcal{L}_{\text{data}}(\hat{x}, \{t_d, y_d\}_{\text{mb}})$ according to (1)
    **else**
        **if** $i = i_{\text{EBM}}$ **then** initialize EBM
        Calculate data loss $\mathcal{L}_{\text{data}}^{\text{EBM}}\left(\{y_d - \hat{x}(t_d)\}_{\text{mb}}, \hat{h}\right)$ from (7)
    **end**
    **if** $i >= i_{\text{EBM}}$ **then** $\omega' = \omega$ **else** $\omega' = 1$
    Calculate PDE loss $\mathcal{L}_{\text{PDE}}(\mathcal{F}, \hat{x}, \{t_c\}_{\text{mb}})$ according to (2)
    Compute total loss $\mathcal{L}_{\text{tot}} = \mathcal{L}_{\text{data}} + \omega'\mathcal{L}_{\text{PDE}}$
    Calculate gradient $\nabla_\theta \mathcal{L}_{\text{tot}}$ and update $\theta = \{\theta_{\text{PINN}}, \theta_{\text{EBM}}, \lambda\}$

---

To this end, we train an EBM and employ it to obtain the negative log-likelihood of the data, given both our models. For the training procedure, we then utilize the following loss function:

$$\mathcal{L}_{\text{tot}} = \mathcal{L}_{\text{data}}^{\text{EBM}}(\{y_d - \hat{x}(t_d)\}_{\text{mb}}, \hat{h}|\theta_{\text{PINN}}, \theta_{\text{EBM}}) + \omega\mathcal{L}_{\text{PDE}}(\mathcal{F}, \hat{x}, \{t_c\}_{\text{mb}}|\theta_{\text{PINN}}, \lambda), \qquad (7)$$

where $\mathcal{L}_{\text{data}}^{\text{EBM}}(\cdot, \hat{h}) = \frac{1}{N_d'}\text{NLL}(\cdot, \hat{h}) = \log Z(\hat{h}) - \frac{1}{N_d'}\sum_{i=1}^{N_d'}\hat{h}(\cdot)$, utilizing (4); note that $\hat{h}$ and hence the NLL are now independent of $t$, because we consider homogeneous noise. Since we do not know the actual magnitude of the noise at each data point, we use the residuals $y_d - \hat{x}(t_d)$ between the current PINN prediction and the measurements as our best guess. These estimates of the noise values then serve as training data for the EBM. In other words, we employ an unconditional EBM to model the PDF of the residuals. With the loss (7), both models are trained jointly until convergence.

### 4.4 TRAINING PINN AND EBM JOINTLY

In Algorithm 1, the training procedure is summarized. Since both PINN and EBM need to be trained in parallel, the optimization procedure can be very challenging. In our experiments, it proved advantageous to start with the standard PINN loss in order to obtain a solution close to the data, and to only subsequently, after $i_{\text{EBM}}$ iterations, switch to the EBM loss, in order to fine-tune the solution. Before the EBM loss is used for the first time, we initialize the EBM by training it for $N_{\text{EBM}}$ iterations on the current noise estimates $y_d - \hat{x}(t_d)$, while keeping all parameters except $\theta_{\text{EBM}}$ fixed. Starting directly with the EBM loss also worked, but it often took significantly longer for the algorithm to converge.

Initializing the EBM only later during the training process is advantageous for two reasons: firstly, the least-squares loss is still a sensible first guess, so initializing the EBM on the residuals stemming from the pretrained network will let it start out with a reasonable form of the likelihood instead of a random one. Secondly, it will give us a better idea on the range in which the residuals between PINN prediction and data lie, allowing for a better normalization of the inputs to the EBM and hence more efficient training.

## 5 EXPERIMENTS

In this section, we compare the performance of the standard PINN from Section 3.1 to that of the PINN with offset parameter and the combination of PINN and EBM at hand of multiple experiments. In the following, we will refer to these models as PINN, PINN-off and PINN-EBM.

In our experiments, we consider homogeneous noise in a variety of shapes. Amongst them are standard Gaussian noise $p(x) = \mathcal{N}(x|0, 2.5^2)$, a uniform distribution $p(x) = \mathcal{U}[0, 10]$ and a Gaussian mixture of the form $p(x) = \frac{1}{3}\left(\mathcal{N}(x|0, 2^2) + \mathcal{N}(x|4, 4^2) + \mathcal{N}(x|8, 0.5^2)\right)$. The values given here

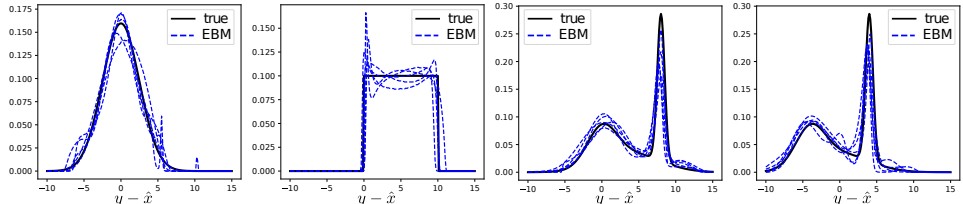

Figure 2: The noise distributions considered in the experiments are depicted. From left to right, we have a Gaussian distribution (G), a uniform distribution (u), a mixture of Gaussians (3G), and the same mixture of Gaussians, shifted to obtain zero mean (3G0). The dashed blue curves give examples of PDFs learned by the EBM during the training process, whereas the black curve gives the true PDF.

are scaled to the example of the exponential function in Section 5.1. For other experiments, the shapes of the curves are retained, but the magnitude of the noise values is rescaled by a factor corresponding to the measurement values of the dataset at hand. In Fig. 2, the different noise distributions are depicted, together with a few examples of the PDFs learned by the EBM during experiments.

When evaluating the performance of our models, the following metrics are considered: the absolute error in the learned values of the PDE parameters ($|\Delta\lambda|$), the root-mean-square error (RMSE) on the validation data, the log-likelihood (logL) of the validation data according to the models, and the square values of the PDE residuals ($f^2$) on the training data. In practice, the log-likelihood is the most relevant performance metric: contrary to the RMSE and $|\Delta\lambda|$, it can also be calculated when the true solution is not known. While $f^2$ can also be calculated without knowledge of the true solution, it only measures how well the learned PDE is fulfilled and not necessarily the correct one.

Further results are discussed in the appendices: learning curves for the experiments are provided in Appendix A. In Appendix B, model performance is investigated as a function of the weighting factor $\omega$ in (6) and (7).

## 5.1 TOY PROBLEM

As a first toy problem, we consider the following simple differential equation,

$$\dot{x}(t) = \lambda x(t), \tag{8}$$

where $\lambda = 0.3$ and with the exponential function, $x(t) = e^{\lambda t}$, as solution.

In Fig. 3, the results for an individual run of PINN, PINN-off, and PINN-EBM are compared; blue dots represent training data and red dots validation data. From this plot, it is clear that PINN converges to a wrong solution, and PINN-off gives only a slightly better prediction, whereas the PINN-EBM result is very close to the true curve.

In Table 1, performance metrics of the different models as obtained for different noise forms are given. The results have been averaged over ten runs with different realizations of the measurement noise.

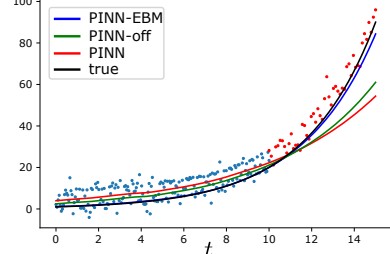

Figure 3: The exponential differential equation (8) with Gaussian mixture noise. Results for one individual run, where the blue dots represent training data and the red dots validation data.

We start by discussing the results for Gaussian mixture noise (3G). It is evident that PINN performs poorly on all metrics. While PINN-off learns the correct parameter $\lambda$ better on average, the variance of its predictions is very high, thereby limiting its use. The PINN-EBM estimate of the parameter is significantly less variable than that of PINN-off, and is very close to the correct value of $\lambda$. In accordance with the quality of the parameter estimates, PINN-EBM clearly outperforms both PINN and PINN-off when considering the RMSE and logL; it achieves the lowest RMSE as well as the highest logL on the validation data.

| noise | | PINN-EBM | PINN-off | PINN |
|---|---|---|---|---|
| 3G | $100|\Delta\lambda|$ | **1.22**±1.12 | 3.96±3.62 | 10.22±1.47 |
| | RMSE | **0.29**±0.29 | 1.07±0.87 | 4.01±0.28 |
| | logL | **-3.48**±0.97 | -7.85±9.0 | -8.88±2.75 |
| | $100f^2$ | **0.51**±0.3 | 30.85±11.51 | 48.83±17.07 |
| u | $100|\Delta\lambda|$ | **1.27**±0.33 | 3.25±2.45 | 12.29±0.66 |
| | RMSE | **0.19**±0.12 | 0.79±0.53 | 5.11±0.17 |
| | logL | **-4.75**±1.14 | -6.49±6.77 | -19.06±3.82 |
| | $100f^2$ | **0.72**±0.29 | 16.25±4.75 | 37.94±12.56 |
| G | $100|\Delta\lambda|$ | 2.45±1.78 | 2.02±1.34 | **1.08**±0.91 |
| | RMSE | 0.6±0.43 | 0.46±0.27 | **0.29**±0.07 |
| | logL | -5.13±2.47 | -5.05±3.55 | **-4.15**±2.48 |
| | $100f^2$ | **0.42**±0.32 | 10.82±5.97 | 11.72±6.58 |
| 3G0 | $100|\Delta\lambda|$ | **1.17**±1.03 | 5.28±3.23 | 2.21±2.03 |
| | RMSE | **0.23**±0.19 | 1.12±0.69 | 0.48±0.15 |
| | logL | **-3.73**±1.29 | -9.76±11.15 | -5.44±3.99 |
| | $100f^2$ | **0.61**±0.41 | 32.2±15.05 | 34.02±15.53 |

Table 1: Results for the exponential differential equation (8) in case of different noise forms (compare Fig. 2). The entries in the table are averages obtained over 10 runs plus-or-minus one standard deviation. Bold font highlights best performance.

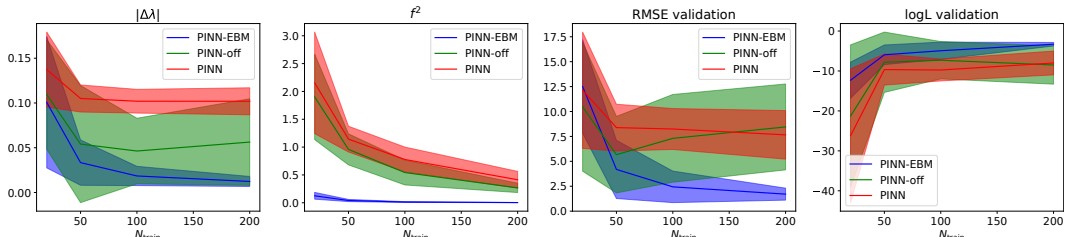

Figure 4: Results for the exponential differential equation (8) with mixture of Gaussians noise as a function of the number of training points $N_{\text{train}}$. Depicted are the means plus-or-minus one standard deviation, obtained over 5 runs.

When considering $f^2$, PINN needs to trade off accuracy in this metric with minimizing the data loss (1), due to the inconsistency between the losses discussed in 4.1. PINN-EBM, on the other hand, does not suffer from this issue and achieves the best fit with the learned PDE.

Three other noise forms are considered in the table: for the uniform noise distribution, the results are very similar to the mixture of Gaussians. In case of Gaussian noise, PINN performs best, which makes sense since it is implicitly using the correct likelihood. PINN-EBM, on the other hand, performs a bit worse as there is some overfit to the noise in the data, compare Fig. 2.

For Gaussian mixture noise with zero mean, where the PDE and data loss are no longer misaligned for the PINN, PINN-EBM still outperforms PINN, although with a lesser margin. This shows that PINN-EBM can successfully mitigate the effects of non-Gaussian noise and significantly improves the quality of the parameter estimates, as well as that of the solution to the regression task. While PINN-off manages to improve upon PINN in the case of non-zero mean noise, the high variance in its predictions makes it less reliable than PINN-EBM.

In Fig. 4, we investigate the impact of the number of training points available. Starting with a small amount of training points $N_{\text{train}} = 20$, all of the models perform similarly. As the number of training points increases, the model performances improve as well, although the improvement beyond $N_{\text{train}} = 50$ seems negligible for PINN. The advantage of more training data is most pronounced for PINN-EBM. This makes sense as more training data will enable the EBM to learn a more accurate

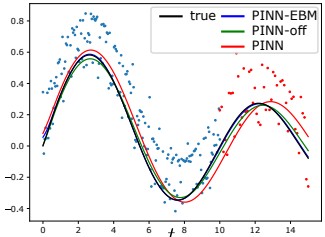

Figure 5: The Bessel equation (9) with Gaussian mixture noise. Results for one individual run, where the blue dots represent training data and the red dots validation data.

| noise | | PINN-EBM | PINN-off | PINN |
|---|---|---|---|---|
| 3G | $100\|\Delta\lambda\|$ | **0.27**±0.22 | 1.81±0.93 | 3.11±1.58 |
| | RMSE | **0.0**±0.0 | 0.03±0.01 | 0.04±0.01 |
| | logL | **0.94**±0.13 | 0.63±0.08 | 0.19±0.32 |
| | $100f^2$ | 0.63±0.14 | 0.39±0.4 | **0.19**±0.05 |

Table 2: Results for the Bessel differential equation (8) in case of mixture of Gaussians (3G) noise. The entries in the table are averages obtained over 10 runs plus-or-minus one standard deviation. Bold font highlights best performance.

representation of the noise distribution. The results for smaller $N_{\mathrm{train}}$ show that the EBM can learn reasonable noise distributions also for smaller data sets; thus, PINN-EBM does not depend on an outsized amount of training data to perform well.

## 5.2 BESSEL EQUATION

To test the framework on a more complicated differential equation, we employ the Bessel equation, a second-order differential equation (Niedziela, 2008; Bowman, 2012),

$$(\lambda t)^2\ddot{x} + \lambda t\dot{x} + ((\lambda t)^2 - \nu^2)x = 0, \tag{9}$$

where we pick $\nu = 1$ and where we have introduced the parameter $\lambda = 0.7$, which is to be estimated by the PINN. Results for one run of this experiment are shown in Fig. 5. When comparing the results obtained here (see Table 2) to those from the previous section, we note the following: PINN-EBM outperforms both PINN and PINN-off on all metrics except $f^2$, and PINN-off performs better than PINN. For this example, the variance of the results for both PINN and PINN-off is smaller than in the previous experiment. One reason for this can be found in the $t^2$ term in (9), which can lead to large values in the PDE loss in (6), thereby putting more weight on PDE fulfillment than on matching the data to a Gaussian.

Another reason why PINN and PINN-off perform better here than for the toy example lies in the shape of the PDE solution: in the previous example from Section 5.1, rather inaccurate estimates of the parameter $\lambda$ could still have been compatible with Gaussian noise of variable strength. Here, on the other hand, the parameter determines the frequency of the oscillation of the curve, leaving less wiggle room for compromise.

## 5.3 NAVIER-STOKES EQUATIONS

For our next experiment, we consider the example of incompressible flow past a circular cylinder and use the dataset from Raissi et al. (2019), to which we add mixture of Gaussians (3G) noise. This setup can be described by the 2D Navier-Stokes equations,

$$f = u_{\mathrm{t}} + \lambda_1(uu_{\mathrm{x}} + vu_{\mathrm{y}}) + p_{\mathrm{x}} - \lambda_2(u_{\mathrm{xx}} + u_{\mathrm{yy}}) \overset{!}{=} 0, \tag{10a}$$

$$g = v_{\mathrm{t}} + \lambda_1(uv_{\mathrm{x}} + vv_{\mathrm{y}}) + p_{\mathrm{y}} - \lambda_2(v_{\mathrm{xx}} + v_{\mathrm{yy}}) \overset{!}{=} 0, \tag{10b}$$

where $u$ and $v$ denote the components of the velocity field and $p$ the pressure; the subscripts indicate derivatives with respect to the different dimensions of the inputs $t = (\mathrm{t}, \mathrm{x}, \mathrm{y})$. Here, the PINN does not output $u$ and $v$ directly, but instead models the potential $\psi$ and the pressure $p$; then $u = \psi_{\mathrm{y}}$ and $v = -\psi_{\mathrm{x}}$. The true values of the parameters are $\lambda_1 = 1$ and $\lambda_2 = 0.01$.

The results are given in Table 3. It is apparent that PINN-EBM learns $\lambda_1$ better than PINN and PINN-off whereas the difference is small for $\lambda_2$. PINN-EBM also performs better in terms of RMSE

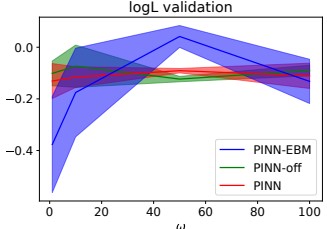

Figure 6: The log-likelihood on the validation data as a function of the parameter $\omega$, weighting the PDE loss. The averages of 5 runs are depicted, plus-or-minus one standard deviation.

| noise | | PINN-EBM | PINN-off | PINN |
|---|---|---|---|---|
| 3G | $100\|\Delta\lambda_1\|$ | **1.19**±0.67 | 2.92±0.47 | 23.1±0.11 |
| | $100\|\Delta\lambda_2\|$ | **0.04**±0.03 | 0.09±0.05 | 0.08±0.06 |
| | RMSE | **0.01**±0.0 | 0.03±0.0 | 0.19±0.0 |
| | logL | **0.03**±0.08 | -0.15±0.07 | -0.4±0.3 |
| | $100f^2$ | **0.04**±0.0 | 0.1±0.01 | 0.18±0.01 |

Table 3: Results for the Navier Stokes equations (10a) in case of mixture of Gaussians (3G) noise. The entries in the table are averages obtained over 5 runs plus-or-minus one standard deviation. Bold font highlights best performance.

and log-likelihood. Here, we have chosen $\omega = 50$ for PINN-EBM since the log-likelihood on the validation data increased when the PDE loss was weighted more strongly (see Fig. 6). For PINN and PINN-off, on the other hand, increasing the parameter did not lead to notable changes in the log-likelihood.

## 5.4 IMPLEMENTATION DETAILS [1]

The datasets employed in Sections 5.1 and 5.2 contained 200 training points, 50 validation points and 2000 collocation points. For the Navier-Stokes example in Section 5.3, 4000 training points, 1000 validation points and 4000 collocation points were used. The collocation points were generated on a uniform grid. For both PINN and EBM, fully-connected neural networks with tanh activation function were used, with 4 layers of width 40 in case of the former, and 3 layers of width 5 for the latter. For the EBM, a dropout layer with factor $0.5$ was inserted before the last layer. Both inputs and outputs of the networks were normalized to their expected ranges. In case of the Navier-Stokes experiment, 5 layers of width 30 were used for the PINN. The EBM was initialized after $i_{\text{EBM}} = 4000$ iterations, except for the Navier-Stokes example, where $i_{\text{EBM}} = 10000$ was chosen. The Adam optimizer with learning rate of 2e-3 was used for both PINN and EBM. The batch size for data points was 200 and the batch size for collocation points was 100. Training the PINN-EBM typically took 30-50% longer than the standard PINN.

## 6 CONCLUSIONS AND FUTURE WORK

In this paper, we demonstrated that the standard PINN fails in case of non-zero mean noise and we proposed the PINN-EBM to resolve this problem; utilizing an EBM to learn the noise distribution allows the PINN to produce good results also in case of non-zero mean and non-Gaussian noise. Using several examples, ranging from a simple toy problem to the complex Navier-Stokes equations, we demonstrated the capabilities of our method and showed that it outperforms the standard PINN by a significant margin in case of non-Gaussian noise. In addition to determining the correct PDE solution, the PINN-EBM also allows for the identification of the true noise distribution, which may result in novel insights into the measurement procedure.

In the future, it would be interesting to investigate combining the PINN-EBM with other improvements to the PINN framework, such as adaptive weighting schemes (McClenny and Braga-Neto, 2020), scheduling approaches (Krishnapriyan et al., 2021; Wang et al., 2022) or additional loss terms (Yu et al., 2022). In principle, our method could also be applied to the problem considered in Bajaj et al. (2021) (see also Section 2): where the standard GP presumably would fail here as well in case of non-Gaussian noise, a representation for the initial conditions could potentially also be learned via PINN-EBM.

---

[1]Code for the project will be made available on Github.

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

# A  LEARNING CURVES

In Figures 7-9, learning curves for the experiments discussed in Sections 5.1-5.3, with mixture of Gaussians (3G) noise, are provided. From the curves, it becomes apparent that PINN-EBM typically converges quickly to the correct solution, after the EBM has been initialized. In case of the exponential differential equation, PINN and PINN-off also converge fast, albeit towards often significantly less accurate solutions. For the Bessel function, both of them converge towards values close to the correct solution but do so only very slowly.

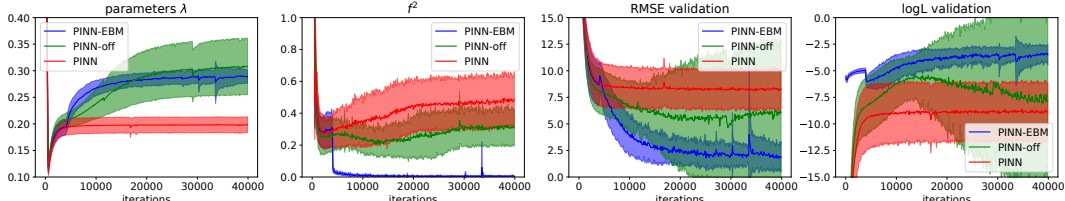

Figure 7: Learning curves for the exponential differential equation (Section 5.1) with Gaussian mixture noise.

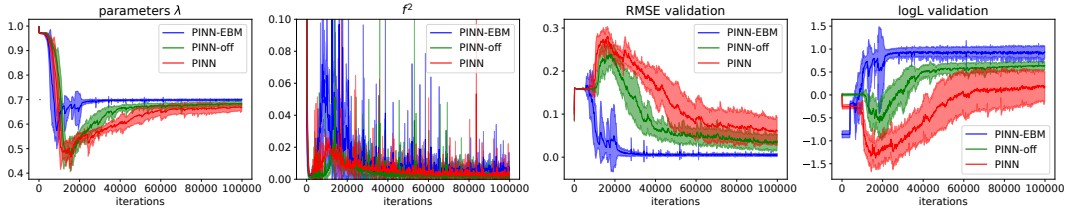

Figure 8: Learning curves for the Bessel equation (Section 5.2) with Gaussian mixture noise.

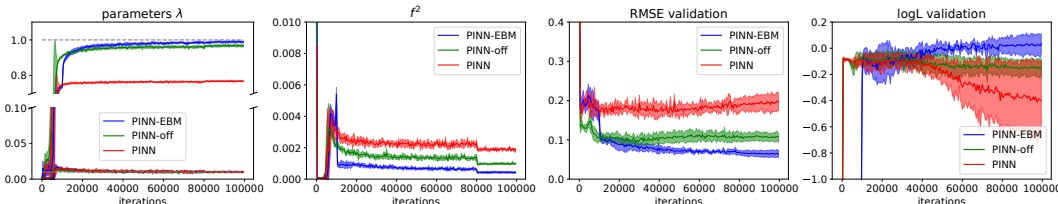

Figure 9: Learning curves for the Navier Stokes equations (Section 5.3) with Gaussian mixture noise.

# B  VARYING THE WEIGHTING FACTOR $\omega$ FOR THE PDE LOSS

In Fig. 10, results for different values of the weighting factor $\omega$ in (6) and (7) are depicted. In case of PINN-EBM, the training starts with $\omega = 1$ and the larger value of $\omega$ is only used as soon as the EBM has been initialized (compare Algorithm 1). The main questions that this investigation is supposed to answer are the following: do small values of the PDE residuals $f^2$ correspond to correct solutions? Is there an optimal value of $\omega$ and how can it be determined?

To answer the first question, it is apparent that $f^2$ becomes smaller as $\omega$ increases. However, the corresponding plots of the RMSE and $\Delta\lambda$ show that this does not imply correctness of the results. In case of the Bessel and the Navier Stokes equations, the results can become significantly worse as $\omega$ increases. This also shows that the PDE loss cannot, in general, override the detrimental effects of the non-Gaussian noise for the standard PINN. One may have assumed that the PDE would only be compatible with a very limited range of PDE parameters and hence for the PINN to be pushed towards the correct solution that way; however, this only seems to be the case for PINN-off when applied to the exponential differential equation.

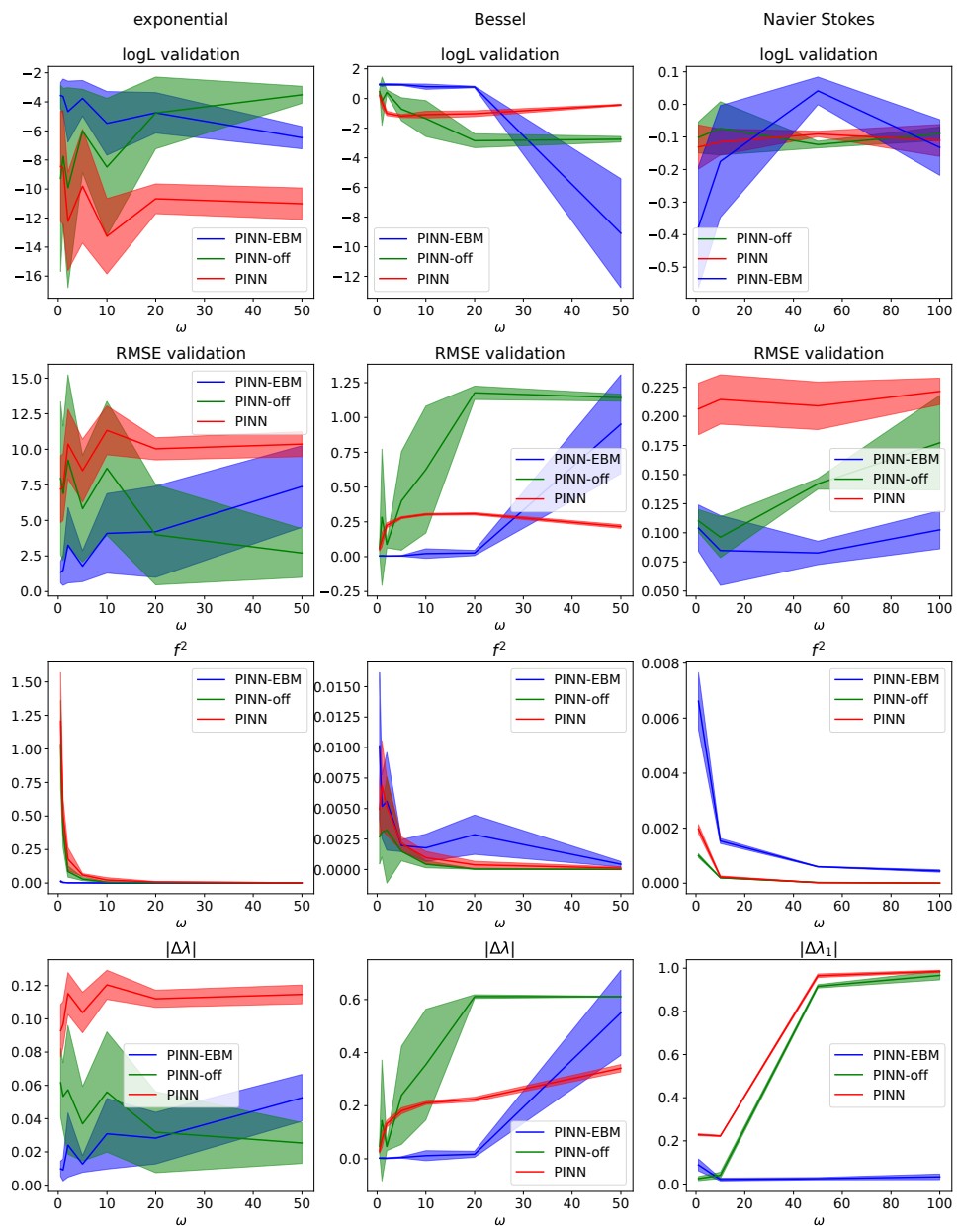

Figure 10: The impact of the weighting factor $\omega$ on model performance is depicted for the different experiments considered in Section 5.

To answer the second question, when considering e.g. the plots for the RMSE, it is apparent that no such value $\omega$ exists. Only PINN seems to consistently perform better for small values $\omega$. To determine the best value of $\omega$ for a given case, the log-likelihood on the validation data remains crucial. For PINN and PINN-off, worse predictions do not always translate into lower values of logL, as the example of the Navier Stokes equations shows. In these cases, it is important to keep an eye on the learned values of $\lambda$ during training, as values $\lambda \approx 0$ can often indicate failure. For the PINN-EBM, however, there is a clear correlation between higher logL and better results. Most importantly, the best solution as learned via PINN-EBM consistently results in higher logL than PINN and PINN-off for any $\omega$.

