# OpenReview forum: "Physics-informed neural networks with unknown measurement noise"
_ICLR.cc/2024/Conference — ICLR 2024 Conference Withdrawn Submission_

### Official Review · Reviewer_99Tt · 2023-10-19

**Soundness:** 3 good
**Presentation:** 2 fair
**Contribution:** 3 good
**Rating:** 6
**Confidence:** 3

**Summary:**

The authors introduce a new training procedure for PINNs which are adapted to unknown measurement noise, i.e., a training procedure which works for any noise model. This is done via EBMs, which are trained jointly with the PINN. Here the EBMs estimate a 1d noise model based on the estimation of the PINN (conditional to the point $t_i$). Since they only estimate a 1d distribution, the (usually intractable) normalization constant can be estimated via numerical integration. The approach is tested on several (partial) differential equations and benchmarked against the standard PINN.

**Strengths:**

The paper is easy to follow (except a few minor points). The idea is interesting and well-executed. The approach outperforms the standard PINN and offset PINN baseline. The experiments are well described, so that I think reproduction should be easy.

**Weaknesses:**

1) While the idea is heuristically clear, it would be interesting whether one can obtain theoretical guarantees. I have got the hunch that it should be possible to cast the framework into one of expectation maximization (EM) algorithms (maybe one slightly needs to change the loss and train alternating instead of jointly). Did the authors give this some thought? This would greatly strengthen the paper in my opinion. For this see e.g. [1]

2) The discussion in 4.1 and 4.2 is a bit confusing. While I think I got the gist of it, please make clear what variables the functions $\mu_{\varepsilon}$ and $\theta_0$ depend on.

3) The metric logL is not clearly defined. How is that calculated in the case of a standard PINN, just Gaussian likelihood?

4) The non-Gaussian noise is a GMM. I would like to see physically more realistic noise models. One thing that could be interesting is whether this approach is able to learn mixed Gaussian noise, i.e., $y = f(t) + \eta_1 + f(t)\ \eta_2$ for normal $\eta_1,\eta_2$ with some variances. While this is still Gaussian, this is a noise model used in practice.

5) Please make the relation to model errors [2] and [3] more clear. Although the model error framework tries to solve a different problem (Bayesian inversion) the ideas are somewhat similar.

6) A very similar is to train a surrogate on the data only (no PINN loss), then estimate the noise via an appropriate model, such as an EBM and then to train the surrogate on a combined loss. Please comment on this.

[1] DeepGEM: Generalized Expectation-Maximization for Blind Inversion, Gao et al

[2] Iterative Updating of Model Error for Bayesian Inversion, Calvetti et al

[3] Noise-aware physics-informed machine learning
for robust PDE discovery, Thanasutives et al

**Questions:**

See weaknesses. I overall like the idea and think it has a lot of merit. A consideration of more realistic noise models and some theoretical guarantees would strenghten the article imo.

---

> ### Author Response · Authors · 2023-11-14
>
> We thank the reviewer for the careful evaluation of our paper and the suggested ways of improving the paper.
>
> To comment on Questions/Weaknesses:
>
> 1) This is an interesting suggestion and we will look into it.
>
> 2) The variable mu_epsilon denotes the mean of the noise distribution. theta_0 denotes a learnable parameter which is supposed to learn mu_epsilon. In the revised version, we will clarify this.
>
> 3) Indeed, in the case of the standard PINN a Gaussian likelihood is employed to estimate the log-likelihood; the parameters of this Gaussian are estimated from the residuals. We will also clarify this in the revised version.
>
> 4) While our method would not allow for this kind of noise in its current form  due to the f(t)*eta_2 term, it should be possible to extend it in a suitable way.
>
> 5) We thank the reviewer for bringing these works to our attention. We will consider them in the revised version.
>
> 6) We would expect the proposed approach to perform worse than PINN-EBM (although probably better than the standard PINN). The reason for this is that a standard neural network (without  the additional regularization provided by the PINN loss) will likely overfit to the noise and in turn yield a worse noise estimate. Furthermore, the offset would still need to be learned jointly with the PINN in the last step, since we have no way of knowing the offset of the learned noise distribution without the PINN-loss.

---

> > ### Comment · Reviewer_99Tt · 2023-11-14
> > **Quick Question**
> >
> > Thanks for the response.
> >
> > Regarding 4) can you explain why your method is not applicable to this setting? I thought the EBM should also be able to learn these kinds of noise models and the EBM is allowed to depend on t?

---

> > > ### Author Response · Authors · 2023-11-15
> > >
> > > The reason is that we consider homogeneous noise - compare equation (7) and the text below. This assumption is necessary in order to obtain residuals that stem from the same distribution and which can in turn be used to train the EBM.
> > >
> > > For noise forms like the one you mentioned, where the noise is nonhomogeneous but for which the relationship between f(t) and y is known, it should still be possible to obtain identically distributed values to train the EBM on. However, this would require some (probably small) alterations to the method.

---

> > > > ### Comment · Reviewer_99Tt · 2023-11-22
> > > > **Post rebuttal**
> > > >
> > > > Overall my opinion of the paper stays the same. I found it to be a good read with room for improvements on the theoretical side as well as more realistic examples.

---

### Official Review · Reviewer_4nHF · 2023-10-28

**Soundness:** 2 fair
**Presentation:** 2 fair
**Contribution:** 2 fair
**Rating:** 3
**Confidence:** 3

**Summary:**

This article proposes a method for training physics informed neural networks (PINNs) when the distribution of measurement noise is unknown. The key idea is to learn noise distribution using an energy-based model on top of training of PINNs. A few numerical experiments show the usefulness of the proposed method.

**Strengths:**

The usefulness of the method is shown by numerical experiments for a few example problems.

**Weaknesses:**

There is little theoretical backing. Extension to high-dimensional and/or non-iid noises would require much heavier computation. Experiments are limited only to synthetic problems.

**Questions:**

Are there any practical problems that could be resolved by the proposed method?

---

> ### Author Response · Authors · 2023-11-14
>
> We thank the reviewer for the evaluation of our paper.
>
> To comment on the stated weaknesses:
>
> *) We would like to point out that we provide theoretical justification for our choice of loss function.
>
> *) While it is correct that we evaluate our method on snythetic data, this is still the case for most PINN research.
>
> To address the question:
>
> *) In the last paragraph of Section 1, we speculate where practical applications of our work may be found.

---

> > ### Comment · Reviewer_4nHF · 2023-11-21
> >
> > > *) We would like to point out that we provide theoretical justification for our choice of loss function.
> > Sorry, I could not find "theoretical justification". For instance, is there any theoretical guarantee for convergence by the choice of loss function?

---

> ### Author Response · Authors · 2023-11-21
>
> We were referring to the discussion in Sections 4.1-4.3, in which we provide justification for our choice of loss function. The main argument is that for our choice of loss function, both terms have the same minimizer in the limit of infinite data. We do not have theoretical guarantees for the convergence.

---

> > ### Comment · Reviewer_4nHF · 2023-11-22
> >
> > Thank you for the explanation. Then, I keep my current score.

---

### Official Review · Reviewer_PpGy · 2023-10-29

**Soundness:** 3 good
**Presentation:** 3 good
**Contribution:** 2 fair
**Rating:** 5
**Confidence:** 4

**Summary:**

The paper proposes the integration of an energy-based model (EBM) to learn the distribution of the noise that is added to samples in a dataset to be modeled by a physics-informed neural network (PINN). A joint loss function is used to train the EBM and the PINN, whereas the EBM can be trained at the same time or with a delayed start with respect to the PINN. Numerical experiments use synthetic data governed by several well-known PDEs from physics, polluted with a variety of noise distributions, to test the performance of the proposed approach.

**Strengths:**

The approach is principled, the description is clear, the results are convincing.

**Weaknesses:**

The proposed approach integrates two well-known models from the literature; the approach is straightforward and the results are not surprising. EBMs have been used before in classification, generative modeling, and regression problems; the authors state that the novelty is in the leveraging of physical knowledge within PINNs. In addition, all the results are focused on synthetic data. Thus the impact of the proposed approach appears limited to the current combination of tools for the usual applications of PINNs.

Minor comments:

In Algorithm 1, within the training loop, i should be updated.

**Questions:**

To better evaluate the impact of the proposed approach, it would be good to discuss the following questions:

(1) How is the formulation of the proposed approach different from the integration of EBM to a regular neural network?

(2) Is there real-world data that would usually be modeled by a PINN where non-Gaussian additive noise is present and for which the proposed approach can be shown to provide better solutions than the baseline PINN?

---

> ### Author Response · Authors · 2023-11-14
>
> We thank the reviewer for the evaluation of our paper.
>
> To answer the questions:
> (1)
> The main differences of combining PINN vs a normal neural network with EBM are the following:
>
> firstly, the PINN loss acts as and additional means of regularization and suppresses overfitting to the noise. The regular neural network, on the other hand, has no knowledge of the PDE and may overfit strongly to noise in the data. The EBM would in turn also learn an incorrect noise distribution that assigns too much weight to smaller noise values.
>
> Secondly, the PINN loss allows for the identification of the noise offset, which would not be possible without the requirement that the solution also adhere to the PDE.
>
> (2)
> In the last paragraph of Section 1, we speculate where practical applications of our work may be found. At the moment we are not aware of a good open source dataset to evaluate our method on.

---

> > ### Comment · Reviewer_PpGy · 2023-11-21
> > **Regarding question 1**
> >
> > I appreciate the answers. For question 1, you are describing the differences between the PINN and a normal neural network. I would like to know more about differences in the integration of the PINN or NN with the EBM - what is the added contribution for  the integration you are proposing? Does the noise offset identification from PINN+EBM not occur from NN+EBM? Can you illustrate?

---

> > > ### Author Response · Authors · 2023-11-21
> > >
> > > The reason why standard NN+EBM could not learn the offset is that it lacks the knowledge on how to 'fix' the noise distribution to the curve learned by the NN. Consider the solution of the exponential equation in Fig. 3: without the PDE constraint, shifting the curve up or down the y axis would merely result in a change of the mean of the noise distribution in the corresponding EBM. Only together with the PDE can the correct curve be identified.

---

### Official Review · Reviewer_pToC · 2023-11-04

**Soundness:** 1 poor
**Presentation:** 1 poor
**Contribution:** 2 fair
**Rating:** 3
**Confidence:** 3

**Summary:**

the paper propose a method to handle measurement noise that has non-zero bias (Eq.7) and algorithm 1. the paper is indeed very hard to read. I would suggest the authors rewrite the paper to allow readers to understand and therefore use this paper for the progress of science. then resubmit the paper in the next conference. I will explain why the paper is hard to read in the next section.

**Strengths:**

A learning method to handle more sophisticated measurement noise.

**Weaknesses:**

I try to help the authors by explaining why the paper is hard to read to me. I hope these feedback can help improve the writing for a future paper.

1. math symbols are not defined when they are first used. examples:

1a. page3, line 3, D_d = {d_d, y_d}. these symbols are not explained and define. y_d was explained only towards end of page 3.

1b. page3, line 3, what is "d"? is this the index of the data point? furthermore D_d is just a set with two elements. how to learn from a set of two elements?

1c. what is the math object of y_d? is it \mathbb{R}^m or \mathbb{R}? t_d \in \mathbb{R}? what is \lambda and what dimension is it?

1d. Eq2. t_c, how to get the colocation points?

1e. algorithm 1, "if i<i_ebm then", what is i?

2. page3 second paragraph. I read this paragraph many times, I still cannot understand it. this paragraph needs to be expanded and writing needs to be clear.

overall the math formulation needs to be improved a lot.

assessment on the results and experiment section becomes invalid if the methods section of the paper is not clear and people cannot reproduce this work.

**Questions:**

see above 'weakness' section.

---

> ### Author Response · Authors · 2023-11-14
>
> We regret that the reviewer found the paper hard to read.
>
> To address the stated weaknesses/questions:
>
> 1a) In the sentence where these quantities are first used, it is stated that this denotes a dataset of N_d noisy measurements of the PINN solution. We will clarify this with the notation \{t_d^i, y_d^i}}_{i=1}^{N_d} and similar for the other sets in the paper.
>
> 1b) The subscript d allows for the distinction between the datasets D_d and D_c, where D_d contains data points (-> d) and D_c contains collocation points (-> c). In the text it is also stated that D_d contains N_d elements and not only two.
>
> 1c) y_d denotes measurements of the PINN solution, which is stated in line 4 on page 3. As such, it has the same dimensionality as x. As is mentioned in the first paragraph of 3.1, x can be multidimensional. Ultimately, the dimension depends on the PDE under consideration. lambda denotes parameters of the differential operator F, which is stated in line 5 on page 3.
>
> 1d) As is stated in the last 2 sentences of paragraph 1 on page 3, collocation points can be chosen arbitrarily in the domain of interest. In section 5 we have explained how it is done for  the different experiments.
>
> 1e) i denotes the loop index. We apologize that we forgot to define it and will clarify this in the revised version.

---

> > ### Comment · Reviewer_pToC · 2023-11-22
> > **keep scores**
> >
> > I like to keep my scores. encourage the authors to keep improving their paper.